# Prophage induction can facilitate the *in vitro* dispersal of multicellular *Streptomyces* structures

**Hoda Jaffal[1], Mounia Kortebi[1], Pauline Misson[2¤ab], Paulo Tavares[1], Malika Ouldali[1], Hervé Leh[1], Sylvie Lautru[1], Virginia S. Lioy[1], François Lecointe[2], Stéphanie G. Bury-Moné[1]** *

1 Université Paris-Saclay, CEA, CNRS, Institute for Integrative Biology of the Cell (I2BC), Gif-sur-Yvette, France, 2 Université Paris-Saclay, INRAE, AgroParisTech, Micalis Institute, Jouy-en-Josas, France

¤a Current address: Medical Research Council London Institute of Medical Sciences, London, United Kingdom
¤ b Current address: Institute of Clinical Sciences, Faculty of Medicine, Imperial College London, London, United Kingdom
* stephanie.bury-mone@i2bc.paris-saclay.fr

**Data Availability Statement:** The RNA-seq data generated during this study have been deposited in the NCBI Gene Expression Omnibus (GEO, https://www.ncbi.nlm.nih.gov/geo/) under the accession

## Abstract

*Streptomyces* are renowned for their prolific production of specialized metabolites with applications in medicine and agriculture. These multicellular bacteria present a sophisticated developmental cycle and play a key role in soil ecology. Little is known about the impact of *Streptomyces* phage on bacterial physiology. In this study, we investigated the conditions governing the expression and production of "Samy", a prophage found in *Streptomyces ambofaciens* ATCC 23877. This siphoprophage is produced simultaneously with the activation of other mobile genetic elements. Remarkably, the presence and production of Samy increases bacterial dispersal under *in vitro* stress conditions. Altogether, this study unveiled a new property of a bacteriophage infection in the context of multicellular aggregate dynamics.

## Introduction

*Streptomyces* are among the most prolific producers of specialized metabolites, with applications in medicine and agriculture [1,2]. These filamentous gram-positive bacteria are widely distributed in the environment and play a key role in soil ecology [3,4]. *Streptomyces* are a rare example of multicellular bacteria capable of forming hyphae with connected compartments. They present a complex life cycle that involves uni- to multicellular transitions, sporulation, metabolic differentiation, programmed cell death, and exploration [3,5].

While *Streptomyces* is the largest prokaryotic genus with over 900 species (Genome Taxonomy Database, Release 08-RS214), relatively few *Streptomyces* phages have been characterized to date. They represent less than 2.5% of the bacterial viruses listed by the International Committee on Taxonomy of Viruses (ICTV) (release 21–221122_MSL37) or the National Center for Biotechnology Information (NCBI) and only 7.5% of the Actinobacteriophage Database

code GSE232795 (https://www.ncbi.nlm.nih.gov/geo/query/acc.cgi?acc=GSE232795). The virome sequencing data are available in the bioproject PRJNA974565 on SRA (https://www.ncbi.nlm.nih.gov/bioproject/974565). We used RNA-seq data available under the accession code GSE162865 (https://www-ncbi-nlm-nih-gov.insb.bib.cnrs.fr/geo/query/acc.cgi?acc=GSE232795). Samy phage complete sequence is available on GenBank under the following accession number: OR263580.1. The Bioproject accession number of this study is PRJEB62744. The numeric data used to generate Figs 1A, 2A–2B, 3A–3B, 4A, 4D, 4F, 4G, S2, S3A–S3C, S4 and S11 are available in S1 Data. The scripts used to generate these figures, along with the script used to run the SARTools DESeq2-based R pipeline, are provided in S2 Data. The SARTools statistical report is included in S3 Data.

**Funding:** This work was supported by the *Agence Nationale pour la Recherche* (ANR-21-CE12-0044-01/STREPTOMICS - https://anr.fr/Projet-ANR-21-CE12-0044 to SBM, HL, SL and VSL). The funding agency had no involvement in the study design, data collection and analysis, decision to publish, or preparation of the manuscript.

**Competing interests:** The authors have declared that no competing interests exist.

**Abbreviations:** CFU, colony-forming unit; DNK, deoxynucleoside monophosphate kinase; eCIS, extracellular contractile injection systems; GO, Gene Ontology; ICE, integrative conjugative element; ICTV, International Committee on Taxonomy of Viruses; IQR, interquartile range; NCBI, National Center for Biotechnology Information; OSMAC, One Strain Many Compounds; PSC, protein supercluster; RDF, Recombination directionality factor; SFM, Soy Flour Mannitol; TEM, transmission electron microscopy; TIR, terminal inverted repeat; VGC, viral genome cluster.

[6]. The *Streptomyces* phages identified so far are double stranded DNA viruses belonging to the *Caudoviricetes* class or *Tectiviridae* family, with the possible exception of an RNA virus detected in a *Streptomyces* transcriptome [7]. Nevertheless, the mechanisms by which bacteriophages recognize, attach to, multiply into, and eventually propagate within multicellular mycelia remain largely unknown. The research on phage diversity and impact on *Streptomyces* physiology and ecology is still in its infancy, with only a few examples to date [8]. Exploring how the multicellular nature and complex differentiation of *Streptomyces* can influence the viral cycle, potentially leading to partial and/or transient resistance to phages, is a promising avenue for research [9,10].

By causing lysis of their host, some *Streptomyces* phages are likely to contribute to the death process of a part of the colony and/or to premature termination of antibiotic production. Accordingly, they can lead to industrial fermentation failure [11]. Moreover, *Streptomyces*–phage interplay may also encompass specific traits linked to the complex life cycle of these bacteria. For instance, some phages can induce the release of specialized metabolites [12,13]. Reciprocally, some bioactive compounds produced by *Streptomyces* can act as antiphage defenses [14–16]. Furthermore, some *Streptomyces* phages encode homologs of sporulation and antibiotic production regulators [17] and can impact the developmental cycle of *Streptomyces* [13]. Reciprocally, susceptibility to phage infection varies along this cycle [13,18].

The last few years have been marked by a sustained effort in the isolation and sequencing of *Streptomyces* phages from environmental samples [12,19–25]. Moreover, the *Streptomyces* genome can constitute *per se* a viral genetic reservoir, through the genomes of temperate prophages that they may host. Indeed, prophage-like sequences are detected in about half of the Actinobacteria [26,27], including 62.4% of *Streptomyces* genomes [26]. While experimental validation is necessary to assess the functionality of these regions, this suggests a significant role for temperate phages in shaping the population dynamics of these bacteria. The prophage state is, perhaps, one of the most important stages in the interaction between phage and host genomes. Indeed, the lysogenic cycle represents an opportunity for phages to confer new properties to their host notably through the expression of moron genes that may contribute to bacterial fitness and environmental niche expansion [28].

In this study, we investigated the (pro)phage biology of *Streptomyces ambofaciens*, a strain valued for its spiramycin production. Although the central chromosome region is considered as a hotspot for integrative element insertion [29], this strain is predicted to harbor a complete prophage in its terminal chromosome compartment. We previously reported that in exponential phase, this compartment is less active transcriptionally compared to the central region and exhibits higher levels of antisense-oriented transcription, especially in the predicted prophage region [30]. Here, we characterize the condition of production of this novel temperate phage from *S. ambofaciens* ATCC 23877, which we named "Samy." This prophage is induced under metabolic stress and activated at the same time as other mobile genetic elements. We observed that this induction promotes bacterial dispersal under *in vitro* stress conditions. This study reports a new property resulting from phage–bacteria interaction in the context of multicellularity.

## Results

### Identification of Samy, a complete prophage, in the *S. ambofaciens* ATCC 23877 chromosome

The comparison of *S. ambofaciens* ATCC 23877 genome to the closely related strain DSM 40697 led to the identification of a genomic island (≈71 kb, 113 genes; **Fig 1A**), which is remarkably large. Indeed, only 12% of *S. ambofaciens* ATCC 23877 genomic islands exceed 45

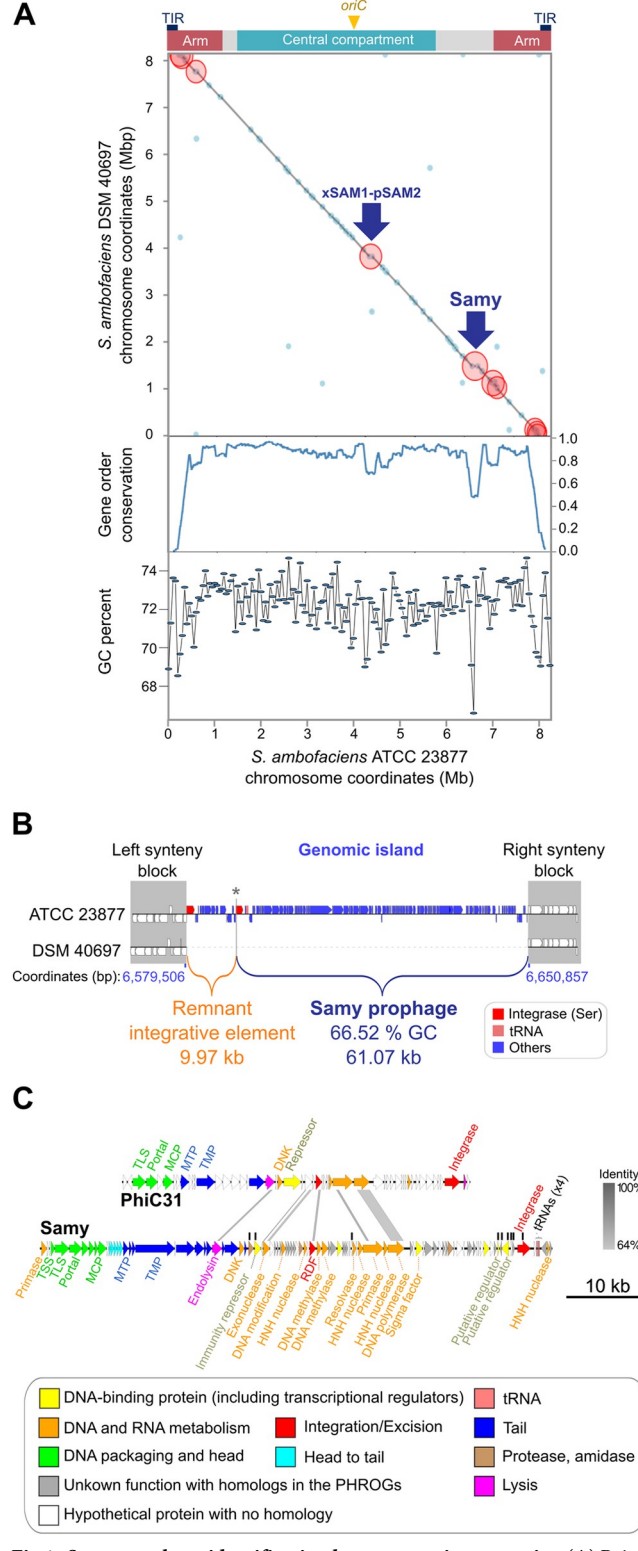

**Fig 1. Samy prophage identification by comparative genomics.** (**A**) Pairwise comparison of *S. ambofaciens* ATCC 23877 and DSM 40697 chromosomes. The gray line indicates the position of genes whose order is perfectly conserved between strains. The blue dots indicate areas of synteny break identified using Synteruptor [32]. The red circles correspond to genomic islands containing at least 20 CDS in one of the strains. The blue arrows indicate the position of xSAM1-pSAM2 integrated elements as well as the genomic island of interest harboring Samy prophage, detailed in

the panels (**B**) and (**C**). The central compartment (delimited by the distal *rrn* operons) [30], the arms (defined as terminal regions devoid of core genes), the terminal inverted repeats (TIRs), and the position of the origin of replication (*oriC*) in *S. ambofaciens* ATCC 23877 chromosome are indicated. The level of gene order conservation between both strains as well as the GC percent content calculated in a window of 50 kb are indicated below the dot-plot. The original plot generated by Synteruptor [32] is available on the "S_ambofaciens_close2" database (https://bioi2.i2bc.paris-saclay.fr/synteruptor/explore_db.php). The data and scripts underlying the lower panel can be found in S1 and S2 Data, respectively. (**B**) Schematic representation of the genomic island containing the Samy prophage. The regions in which gene order is perfectly conserved in *S. ambofaciens* ATCC 23877 and DSM 40697 chromosomes are defined as the left and right synteny blocks (gray). The coordinates of the genomic island borders in *S. ambofaciens* ATCC 23877 strain are indicated in blue. The genomic island is composed of 2 regions: a remnant integrative element and the Samy prophage. These regions are separated by a short intergenic region (represented by an asterisk; 305 bp located from 6,589,473 to 6,589,777 bp) present in both strains. Two serine integrase coding sequences (red) have been predicted. The prophage also contains 4 tRNA encoding genes (pink). (**C**) Annotation of Samy sequence and comparison with the genome of the phylogenetically related PhiC31 phage. Gene functions are color-coded as detailed in the legend. The annotation of PhiC31 (Lomovskayavirus C31) genes was previously reported in [33] and [34]. Black vertical lines indicate the nine Samy genes identified as overexpressed compared to the entire Samy phage genome across all non- or poorly induced conditions (S3 Table). The genome comparison was performed using Easyfig software (e-value $<10^{-3}$). The percentage identity between DNA homologous sequences in Samy and PhiC31 is shown in shades of gray. Other abbreviations: DNK, Deoxynucleoside monophosphate kinase; MCP, Major capsid protein; MTP, Major tail protein; RDF, Recombination directionality factor; TSS, Terminal small subunit; TLS, Terminal large subunit; TMP, Tape measure protein.

genes [30]. This region exhibits the lowest guanosine and cytosine (GC) content (≈66%) and is predicted to contain a complete prophage (**Fig 1A**), as previously reported [26,30]. This latter, further named "Samy", is exclusively present in the ATCC 23877 strain and stands as the sole complete prophage identified in its genome. The ATCC 23877 and DSM 40697 strains share 99.04% of average nucleotide identity calculated by using the BLASTn algorithm (ANIb) [31], indicating that they belong to the same species. The absence of Samy in the DSM 40697 strain indicates that the infection is relatively recent, i.e., occurred after speciation. The identification of a direct repeat ("TCGGGTGTGTCG") in front of a serine integrase gene and approximately 61 kb downstream prompted us to propose this sequence as belonging to the left (*attL*) and right (*attR*) attachment sites of the Samy prophage (6,589,778 to 6,650,857 bp position in the *S. ambofaciens* ATCC 23877 chromosome). Immediately upstream, a region of approximately 10 kb contains only a few phage genes, including one coding for a putative integrase, indicating that it may correspond to a remnant prophage (**Fig 1B**). The mosaic composition of the genomic island suggests that it is located in a region prone to integration and/or fixation of exogenous sequences.

Samy prophage presents a typical GC content (66.53%) and size (61.07 kb, 102 genes) compared to other *Streptomyces* phages (**S1 Fig** and **S1 Table**). Annotation of Samy sequence allowed identification of most structural genes that are essential to form a complete phage (for instance, encoding head, neck and tail proteins, base plate and tail fiber proteins) as well as replication proteins, endolysin, several nucleases, transcriptional regulators, integrase, and recombination directionality factor (**Fig 1C** and **S2 Table**). Altogether, these analyses suggested that this prophage may be able to produce viral particles. This led us to investigate the conditions for its expression.

## Activation of Samy prophage and other mobile genetic element expression is correlated with a general stress response

The linear chromosome of *S. ambofaciens* ATCC23877 is delimited by distal ribosomal RNA (*rrn*) operons into distinct spatial compartments [30,31]. The terminal compartments are enriched in poorly conserved sequences and exhibit lower transcriptional activity compared to the rest of the chromosome [30,31]. The expression of these variable regions is generally conditional, i.e., induced under specific conditions such as the stationary growth phase in MP5

medium for the cluster encoding congocidine antibiotic biosynthesis [35]. An empirical approach is often required to characterize the inducing conditions such as the OSMAC ("One Strain Many Compounds") approach, based on strain cultivation in different environmental conditions [36]. We used a similar approach by analyzing the Samy transcriptome in 13 growth conditions including routine *Streptomyces* complex or minimal media (**S6 Table**), liquid or solid conditions, and different time points (**Figs 2** and **S2A**). We observed 3 conditions of 30 h growth on plates (named "HT," "ONA," or "NAG") in which Samy prophage is globally expressed, suggesting that the lytic cycle of Samy was induced. Its expression level was the strongest in "HT" condition (**Fig 2A**). HT medium was developed by Hickey and Tresner [37] to induce sporulation of *Streptomyces* strains that sporulate poorly on growth media classically used for this purpose. Interestingly, HT condition is also associated with strong expression of several sequences from mobile genetic elements: the pSAM1 plasmid, which was predicted to contain a prophage-like sequence [26] (**Figs 2B and S3A**); the integrative conjugative element (ICE) pSAM2 (**Figs 1A, 2B and S3B**) as well as several other genes of phage origin such as 2 orthologs of phage tail-like nanostructures previously identified as extracellular contractile injection systems (eCIS) [38,39] (SAM23877_RS16255, SAM23877_RS16260); and 2 genes outside a proviral context (the phage tail protein SAM23877_RS23520, and the phage holin family protein SAM23877_RS09270). We confirmed that Samy phage was produced in HT liquid medium ($\approx 10^6$ phages/ml) but not MP5 medium by performing qPCR (**Fig 3A**).

We analyzed Samy transcriptomes in non- or poorly inducing conditions (>38% of Samy genes remaining in the lowest "CAT_0" category, i.e., all conditions except "HT," "ONA," and "NAG"). In these conditions, the Samy transcriptome presents a bimodal pattern, as previously described for other phages [40] (**S4 Fig**). The most transcribed genes in non- or poorly inducing conditions could be involved in the lysogenic cycle of Samy or could be morons, i.e., genes involved in the fitness of the bacteria under these growth conditions or in their resistance to other phages (**Fig 2A and S3 Table**). Among them, the gene we annotated as a putative immunity repressor (SAMYPH_32/SAM23877_6125) was highly expressed under all conditions tested, reinforcing our prediction.

We determined host transcriptome characteristics during Samy induction by comparing the 3 conditions associated with Samy induction ("HT," "ONA," and "NAG") to the closest control condition, lacking Samy induction (**S2B and S2C Fig**). This latter corresponds to 30 h growth on mannitol minimal medium plate ("MM" condition). Gene Ontology (GO) enrichment and DESeq2 analyses revealed that Samy is induced in conditions associated with a strong repression of genes related to nitrogen assimilation (**S2 Table**). Several genes linked to adaptation to osmotic and oxidative stresses are also co-up-regulated with Samy. They notably encode gas vesicles, ectoine and ergothioneine biosynthesis processes (**S2 Table**). Moreover, several sigma factors are differentially expressed in the "HT" condition (compared to the "MM" condition): Six are induced including SigE, a regulator of cell envelope integrity [41], and SigF, which is required for sporulation [42]. Ten sigma factors are repressed, including the principal sigma factor SigA (**S2 Table**). Altogether, these data indicate that the "HT" condition is associated with a profound transcriptional change, probably reflecting a stressful situation.

In parallel, the genes in the GO functional categories "translation" and "arginine biosynthetic process" were strongly repressed (adjusted *p*-values, calculated by g.Profiler, of $3.7 \times 10^{-6}$ and $5.6 \times 10^{-6}$, respectively). Indeed, 34 host tRNAs were repressed, whereas all 4 of Samy's tRNAs were induced. This illustrates how tRNA expression can be part of phage strategy in a translation inhibiting situation.

The SOS response is classically associated with the activation of prophages and has been described in *Streptomyces venezuelae* [43]. In *S. ambofaciens* ATCC 23877, transcripts of the key actors of the SOS response such as RecA, LexA, RecN, DnaE, RuvA Holliday junction

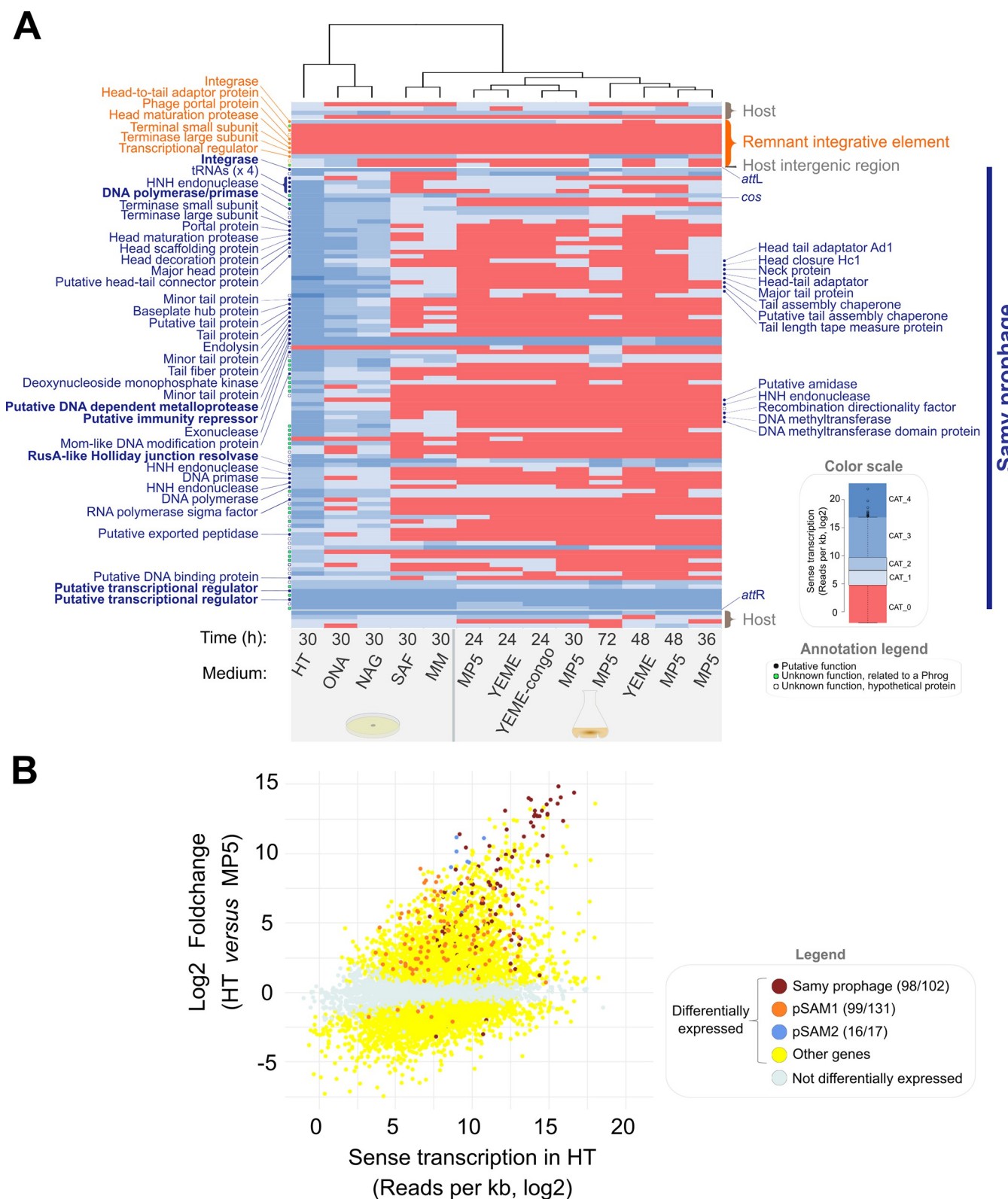

**Fig 2. Massive awakening of mobile genetic elements in HT medium.** (**A**) Heatmap of Samy region transcription in different growth conditions. Normalized RNA-seq data (**S2 Table**) were categorized in a color scale reflecting the expression of each gene relative to whole genome expression in each condition, as indicated in the Annotation legend. Each line represents a gene, ranked according to their order on the genome. For the remnant integrative element as well as

for Samy prophage, a prediction of the gene product is indicated when available. Growth conditions (columns) are ranked by hierarchical clustering according to Samy transcriptome profiles. The growth conditions are detailed in **S2A Fig** and **S6 Table**. Phage genes of predicted function that stand out as overexpressed compared to the entire Samy phage genome across at least 6 non- or poorly induced conditions (**S3 Table**) are written in bold. The data underlying this panel can be found in **S2 Table** and **S1 Data**. (**B**) Plot of the log ratio of differential expression as a function of gene expression after 30 h growth in HT solid medium compared to 24 h growth in liquid MP5 medium. The color of differentially expressed genes (adjusted $p$-value $< 0.05$) is indicated. The number of regulated genes relative to the total number of genes is shown in brackets for the mobile genetic elements of interest. The data and scripts underlying this panel can be found in **S1** and **S2 Data**, respectively.

DNA helicase [43] were not up-regulated in the "HT" condition compared to the "MM" reference (**S2 Table**). This result suggests that the SOS response is not involved in the induction of Samy production in the "HT" condition or may be so transient and/or only present in part of the cell population that it was not detected by our bulk transcriptome analysis. Moreover, addition of mitomycin C DNA crosslinker and ciprofloxacin topoisomerase poison in MP5 medium, at the minimal concentrations that inhibit bacterial growth (1 and 2 μg/mL, respectively), did not induce Samy production measured by qPCR at 30 min, 2 h, and 24 h (Ct $> 28$, 3 independent experiments). This observation reinforces the hypothesis that the SOS response may not be involved in the induction of Samy, at least under our study conditions.

To conclude, in our study, the activation of Samy and other genetic elements is correlated with huge physiological changes associated notably with nitrogen imbalance and translation inhibition and a general stress response.

## Samy production increases in alkaline conditions

In order to determine the key compounds responsible for Samy induction, and to set up monitoring of its production in liquid medium, we tested variants of the HT liquid medium by eliminating one or more of its constituents. The simplest version of a medium associated with high viral titers, as quantified by qPCR, was named "BM" (for "Bacteriophage production Medium") (**Fig 3A**). We noted a fairly high inter-experimental variability in Samy titer in BM medium ($8.10^6$ up to $2.10^{10}$ phage/ml, depending on the experiment), which might reflect its stochastic induction within the cell population. However, production kinetics is the same in all experiments: Samy particles are released between 24 and 48 h of growth in BM medium, the titer being almost constant over the following days (**Fig 3B**).

Interestingly, we observed a positive correlation between viral titer and pH, which is statistically significant (Spearman rank correlation rho = 0.73, $p$-value = 0.016). Indeed, adding buffering compounds (for instance, MOPS or dextrin) in BM medium strongly decreased phage production (**Fig 3A**). These results indicate that alkalinity constitutes an additional signal triggering Samy production and/or increasing virion stability in the supernatant.

## Samy is a new active siphophage

Transmission electron microscopy (TEM) revealed that Samy is a siphophage with a capsid diameter of 65.4 ± 2.8 nm ($n = 35$), and a long rather rigid tail 248.7 ± 7.4 nm in length ($n = 32$) and 11.7 ± 0.4 nm thick ($n = 12$) (**Figs 3C, S5A and S5B**). Remarkably, Samy virions are copurified with eCIS (**S5A and S5B Fig**) consistently, with the transcription of the orthologous genes encoding this system (SAM23877_RS16255, SAM23877_RS16260) in the "HT" condition (**S2 Table**).

High-throughput sequencing of these CsCl-purified particles revealed that Samy phage is the only double-stranded DNA phage produced in this condition. In total, 8,671,900 (99.5%) out of 8,712,953 reads successfully aligned to the *S. ambofaciens* ATCC 23877 genome corresponded to Samy. The remaining reads mapped all along the *S. ambofaciens* genome and may correspond to contaminating bacterial DNA and/or transduced DNA. Interestingly, almost no

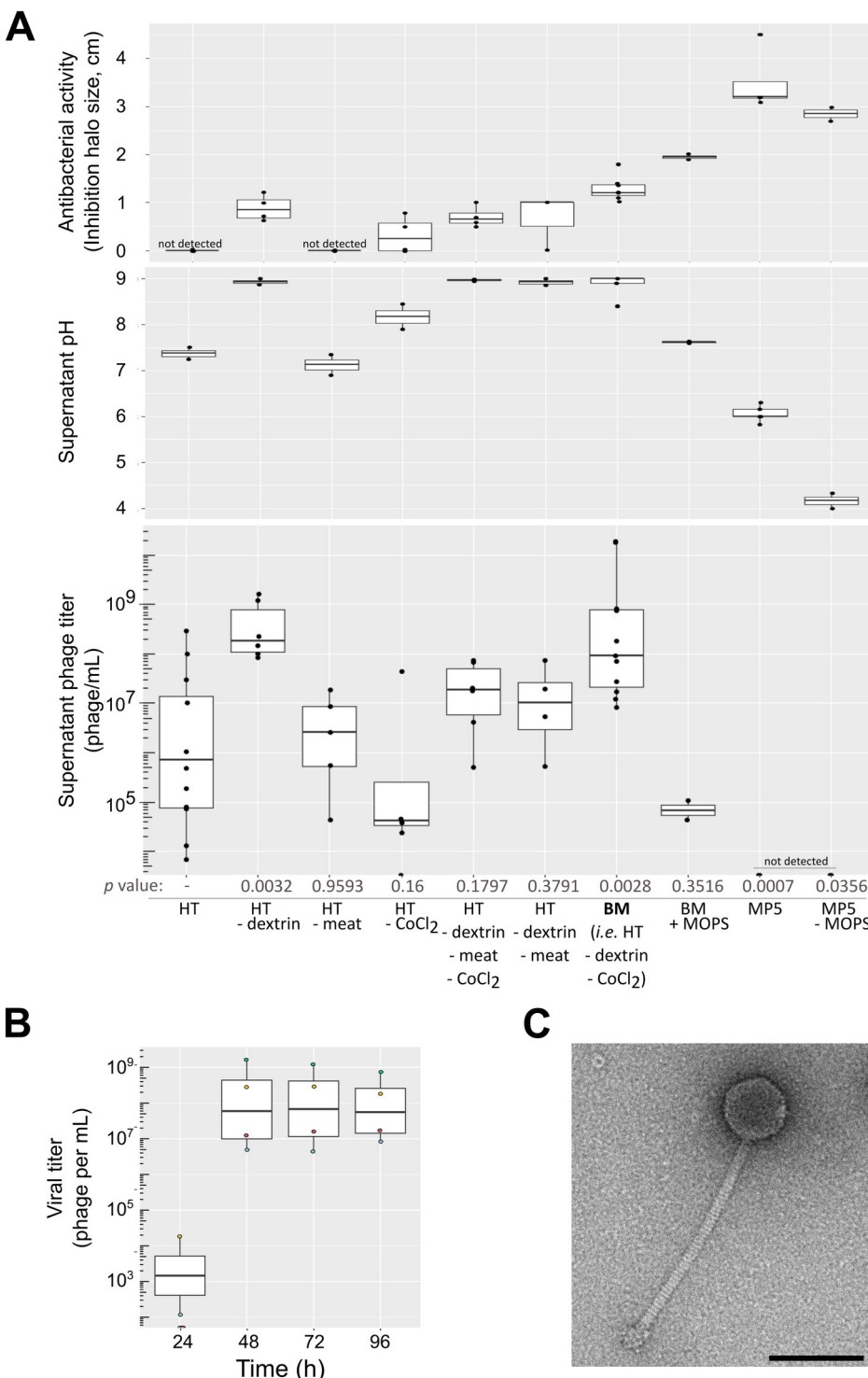

**Fig 3. Samy phage production and morphology.** (**A**) Impact of medium composition on Samy phage production, final pH, and antibacterial activity of *S. ambofaciens* ATCC 23877 supernatants. The supernatants of *S. ambofaciens* ATCC 23877 grown in different media (composition summarized below the graph and detailed in **S6 Table**) were harvested after 4 d and 0.2 µm-filtered. Phage titer was determined by qPCR after DNAse treatment. The pH of the medium and the antibacterial activity against *Micrococcus luteus* of the supernatant after 4 d of growth are indicated on the top. All media had an initial pH of 7.3 (±0.1), except MP5 and MP5 devoid of MOPS, which had a pH of 7.5 (±0.1). All the boxplots represent the first quartile, median, and third quartile. The upper whisker extends from the hinge to the largest value no further than 1.5 * the interquartile range (IQR, i.e., distance between the first and third quartiles).

The lower whisker extends from the hinge to the smallest value at most 1.5 * IQR of the hinge. Each dot represents an independent experiment. The p-values of two-sided Wilcoxon rank sum tests with continuity correction is indicated for each comparison to the viral titer observed in HT condition. The data and scripts underlying this panel can be found in **S1** and **S2 Data**, respectively. (**B**) Kinetics of phage production. *S. ambofaciens* ATCC 23877 was grown in BM medium. The viral titer of supernatants filtered and DNAse-treated were determined by qPCR. Each color represents an independent experiment. The boxplot is plotted as described in the legend of the panel (**A**). The p-value of two-sided Wilcoxon rank sum tests with continuity correction is 0.0294 when comparing the viral titers at 24 h and 48 h, and 1 (no difference) when comparing the evolution of the titer on the following days. The data and scripts underlying this panel can be found in **S1** and **S2 Data**, respectively. (**C**) Imaging of Samy phage produced by transmission electron microscopy. *S. ambofaciens* ATCC 23877 was grown during 4 d in BM medium. The supernatant was concentrated by CsCl-gradient ultracentrifugation. Viral particles were negatively stained with uranyl acetate (also see **S5A and S5B Fig**). Scale bar: 100 nm.

read (<0.01%) mapped on the pSAM1 suggesting that its predicted phage is defective and/or not induced in BM medium. The analysis of sequence coverage at termini positions allowed the prediction of the Samy cohesive *cos* sequence ("CGTTAAGGTGC"; **S6 Fig**). Reads confirmed the borders of the prophage and did not show any trace of lateral transduction, consistent with a *cos* packaging mechanism (**S6 Fig**).

The complete nucleotide sequence of Samy exhibits no substantial similarity to any other deposited phages using BLASTn (with more than 70% nucleotide identity across the genome), indicating its classification as a new phage genus. Accordingly, Samy stands as a singleton in the Actinobacteriophage Database (https://phagesdb.org/phages/Samy/), representing the exclusive member of a distinct cluster identified in a comprehensive analysis of actinobacterial prophages conducted by Sharma and colleagues [26]. To delve deeper, we employed specialized tools for clustering phages by comparing their entire proteomes. A proteomic tree created by ViPTree placed Samy within a cluster primarily consisting of *Streptomyces* and *Arthrobacter* phages, albeit on a separate branch (**S7A Fig**; similarity scores $S_G < 0.05$). The 32 actinophages within this group underwent further analysis using VirClust, resulting in 29, including Samy and PhiC31, clustered together into a viral genome cluster (VGC; **S7B Fig**). Phages within this VGC shared 4 protein superclusters (PSCs), corresponding to the Recombination directionality factor (RDF; excisionase), deoxynucleoside monophosphate kinase (DNK), DNA primase, and DNA polymerase. The VirClust parameters applied here followed recommendations for delineating tailed phage taxa at the family level [44]. However, Samy exhibited a lower number of shared PSCs compared to other phages within the VGC, and its silhouette width value approaching 0 introduces uncertainty regarding its precise taxonomic classification at this time.

Additionally, Samy virions are infectious, since we observed lysis using *Streptomyces lividans* TK24 and *S. ambofaciens* ATCC 23877 strains as hosts (**S5C and S5D Fig**) but not 4 other strains listed in **S4 Table**. We noticed variations in Samy infection efficiency across procedures and experiments, along with the necessity to concentrate the viral particles through overnight centrifugation for plaque observation. These observations suggest either low infectivity, low burst size, and/or challenges in plaque detection due to the phage's temperate nature.

Altogether, these results indicate that Samy is an active prophage that encodes a complete and infectious siphophage belonging to a novel genus.

## Impact of Samy phage on *Streptomyces* growth and antibiotic production

We first compared ATCC 23877 (Samy$^+$) and DSM 40697 (Samy$^-$) strain growth in BM medium, using MP5 medium as a control. Surprisingly, the exponential phase of growth started about 8 h earlier in BM than in MP5 medium, suggesting that germination was more efficient or rapid in this condition (**Fig 4A**). At later stages of growth, the increase in biomass

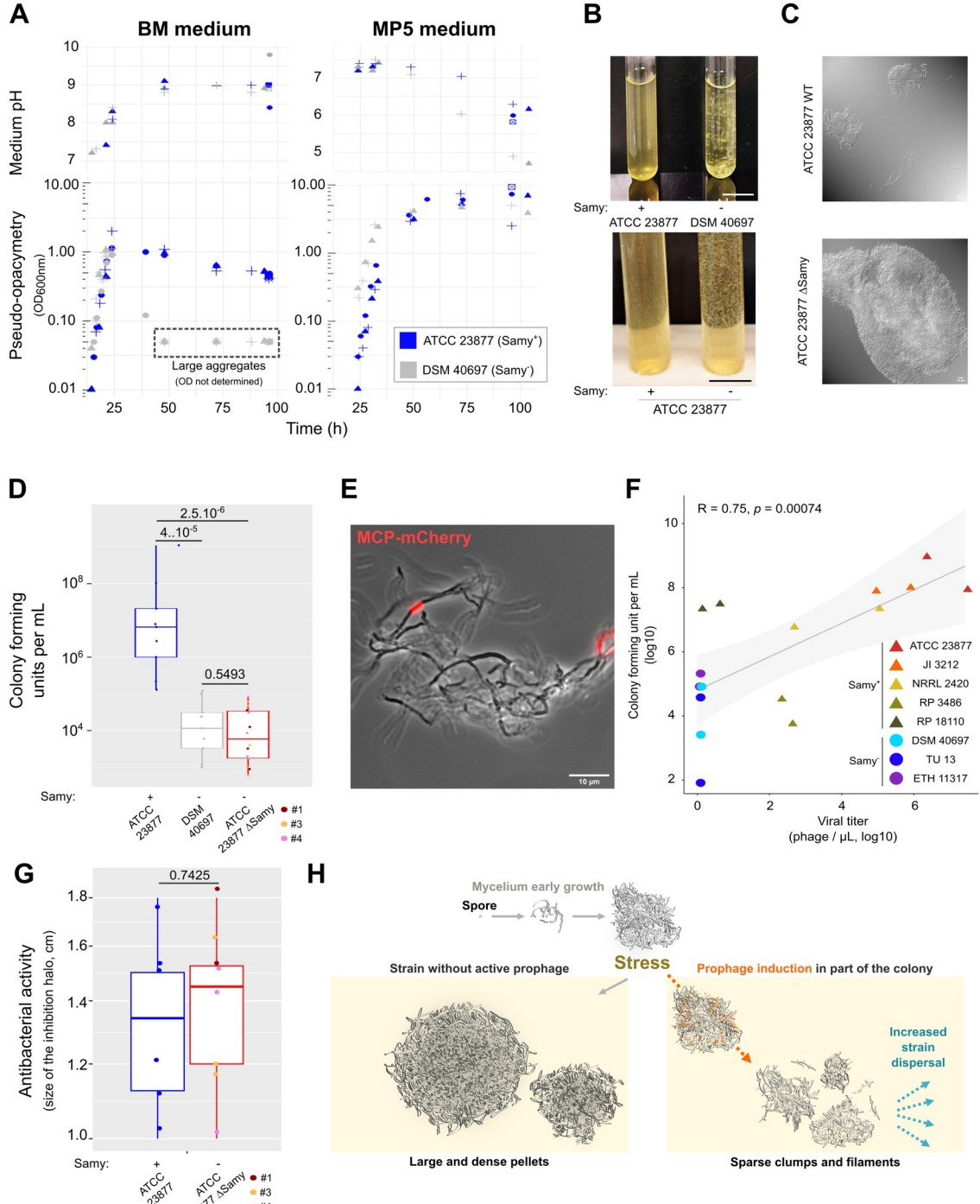

**Fig 4. Dispersion phenotype associated with the production of the Samy phage.** (**A**) Growth and pH in BM and MP5 media. Each symbol represents an independent experiment. Due to the presence of large aggregates (see panel **B**), the pseudo-opacimetry (OD$_{600\ nm}$) was not exploitable for DSM 40697 strain after 48 h of growth in BM medium. (**B**) Dispersed *versus* aggregated growth of *S. ambofaciens* strains in BM media. The results are representative of the appearance of the most frequently observed cultures, as the quantity and size of cell clusters may vary from one experiment to the other. The *S. ambofaciens* ATCC 23877 Δ*Samy* strain corresponds to clone #3, deleted from at least the phage integrase by a CRISPR-based approach (**S4 Table and S8 Fig**). Scale bar: 1.5 cm. (**C**) Microscopy of *Streptomyces* colonies after 4 d growth in BM medium. *S. ambofaciens* ATCC 23877 and its derivate CRISPR-deleted of Samy prophage (clone #3) were grown in BM medium and imaged using differential interference contrast microscopy (also see **S9A Fig**). Scale bar: 10 μm. (**D**) Colony-forming units after 4 d of growth in BM medium. The *S. ambofaciens* ATCC 23877 Δ*Samy* strains correspond to 3 independent mutants (#1, #3, and #4) we obtained by a CRIPSR-based approach (**S4 Table**). The

boxplot is plotted as described in the legend of **Fig 3A**. Each dot represents an independent experiment per condition. The *p*-values of two-sided Wilcoxon rank sum tests with continuity correction are indicated for each comparison. Please note that this number is the result of both cell survival and mycelium dispersal (propensity to form separate colonies). (**E**) Pattern of Samy MCP-mCherry expression in a *S. ambofaciens* colony. Bacteria encoding a MCP-mCherry fusion were inoculated from plates and grown 4 d in BM medium before red fluorescence imaging. The overlay of red fluorescence and bright field images is shown. Additional images and controls are presented in **S9C Fig**. Scale bar: 10 μm. (**F**) Correlation between Samy phage production and colony-forming units by diverse *S. ambofaciens* strains. *S. ambofaciens* strains obtained from distinct collections (**S4 Table**) were grown during 4 d in BM medium, before counting colony-forming units and phage production in the supernatants. Even strains ATCC 23877 and DSM 40697 were cultured for this experiment from stocks produced by another laboratory. Each dot represents an independent experiment per strain. The correlation was analyzed by a Spearman rank correlation test. Results of 2 independent experiments. (**G**) Bioassay performed with the supernatants of *S. ambofaciens* ATCC 23877 WT strain and its derivatives deleted in Samy regions (clones #1, #3, and #4; **S4 Table** and **S8 Fig**). Then 50 μL of filtered supernatant from *S. ambofaciens* culture after 4 d in BM medium was deposited on a *Micrococcus luteus* mat, as previously described [30]. The size of the *M. luteus* growth inhibition halo after 24 h incubation at 37°C was measured. The boxplot is plotted as described in the legend of **Fig 3A**. The *p*-values of two-sided Wilcoxon rank sum tests with continuity correction are indicated. (**H**) Model of the stress- and phage-induced dispersed growth of *Streptomyces* in liquid medium. After germination of the spore, the multicellular bacteria form successively primary and secondary mycelia. In the absence of prophage, the *S. ambofaciens* DSM 40497 tends to form particularly large and dense pellets in response to metabolic stress encountered in BM medium (nitrogen unbalance, translation inhibition, basic pH). Under the same conditions, phage production by *S. ambofaciens* ATCC 23877 is correlated with dispersed growth in sparse clumps and filaments. We propose that phage-induced cell death within the secondary mycelium leads to a dislocation of multicellular aggregates. The subsequent increase in the number of small colonies promotes strain dispersal under stressful conditions. The data and scripts underlying the A, D, F, and G panels can be found in **S1** and **S2 Data**, respectively.

stopped in BM medium quite early (at a pseudo-opacimetry of about 1), whereas it reached higher densities in MP5 medium (**Fig 4A**). We found that this cessation of growth in BM medium was correlated with the alkalinity of the medium (**Fig 4A**). In addition, both strains presented different phenotypes in the following days of growth in BM medium: While the strain ATCC 23877 remained relatively dispersed at a density that did not drop much, strain DSM 40697 formed large pellets (**Fig 4B**).

We used a CRIPSR approach to obtain ATCC 23877 derivate deleted for part or all of the prophage (**S4 Table** and **S8 Fig**). Although the 3 clones tested carry different Samy deletions, they all presented the same phenotype. After 4 d of growth, aggregates appeared in the cultures, but not in the parental strain (**Fig 4B**). Microscopic observations of the cultures confirmed the formation of larger and denser cell pellets in the absence of the prophage (**Figs 4C and S9A**). Interestingly, the formation of cell aggregates during growth in BM medium was also observed with *Streptomyces coelicolor* A(3)2, another *Streptomyces* species that does not contain Samy prophage (**S9B Fig**). The DSM 40697 strain, however, exhibits larger aggregates in BM medium compared to the ATCC strain devoid of the complete Samy prophage, suggesting that prophage induction is only one of the factors influencing cluster size under these specific conditions.

Because of the multicellular nature of *Streptomyces*, the colony-forming units (CFUs) count reflects not only cell survival but also the number of multicellular clusters *per se* regardless of the number and proportion of living or dying cells in the clusters. Thus, we hypothesized that Samy induction could modify the number of culturable and viable colonies in the medium. We determined the number of CFUs after 4 d growth in BM medium of the strains ATCC 23877 (Samy$^+$), DSM 40697 (Samy$^-$) and ATCC 23877 Δ*Samy*. To optimize bacterial growth after stressing conditions, counting were carried out on Soy Flour Mannitol (SFM) medium, widely used for routine cultivation of *Streptomyces* species since it promotes the sporulation of most of them [45]. Remarkably, the number of CFUs was around 2-logs higher in the phage-producing ATCC 23877 strain than in its phage-deleted derivative or the DSM 40697 strain (**Fig 4D**). Thus, the activation of Samy prophage was correlated with a drastic increase in the number of viable multicellular clusters formed by the ATCC 23877 parental strain. During a return to a condition more favorable to growth (exemplified here by the SFM medium used to

count the colonies), this property resulted in an increase in the number of CFUs compared to strains having formed larger cell aggregates.

The presence of the Samy prophage analysis within the genomes of "surviving" clones was confirmed by PCR (S10 Fig). This suggests that the colonies that persisted did not arise from bacteria that had lost the prophage, but rather from those that had not entered the lytic cycle. To assess the induction pattern, we engineered a strain that expresses the major capsid protein of Samy fused to mCherry (MCP-mCherry). Red fluorescence was observed in some segments of the filaments (Figs 4E and S9C), suggesting that Samy phage production was localized to certain areas of the colony and retained within specific filament sectors.

To evaluate the potential impact of storage and strain evolution in laboratory conditions on the growth phenotype, we performed a comprehensive analysis on multiple *S. ambofaciens* strains (harboring or not Samy prophage) sourced from diverse collections (S4 Table). Once again, we observed a positive correlation ($R = 0.75$, $p = 0.00074$, Spearman method) between the production of Samy and the number of CFUs following a 4 d growth period in BM medium (Fig 4F).

In order to test whether a soluble factor present in the culture supernatant could contribute to the dispersal phenotype, we added a conditioned supernatant of the parental strain grown 4 d in BM medium to a 24 h culture of the phage-deleted strain. We found no increase in colony number after 4 d of incubation in BM medium (S11 Fig), indicating that the prophage needs to be present within the cells to increase the *in vitro* dispersal.

Finally, we evaluated the impact of Samy phage production on antibacterial activity, a key step in the development cycle of *Streptomyces*. Interestingly, the antibacterial activity of the ATCC 23877 strain in BM condition is not affected by the presence or absence of an intact prophage copy (Fig 4G).

Altogether, these results indicate that the induction and production of Samy prevents the formation of large cell aggregates by *S. ambofaciens* ATCC 23877 in a stressful condition, promoting bacterial dispersal *in vitro*, without causing a massive death of antibiotic-producing cells.

## Discussion

In this study, we describe the conditions of awakening of a *Streptomyces* prophage associated with profound physiological changes within the colony. In HT broth, a medium initially optimized to promote *Streptomyces* sporulation [37], we have described for the first time a massive activation of mobile genetic element expression (Samy prophage, pSAM1 plasmid, and pSAM2 ICE; Figs 2 and S3) and other phage-related genes such as those encoding eCIS [38,39] (S5A Fig and S2 Table), as well as a high-stress situation associated with an imbalance in nitrogen metabolism and translation inhibition (S2 Table).

This phenomenon seems to be concomitant with physiological modifications that may contribute to strain dispersal in this condition. For instance, we observed increased expression of the genes encoding the synthesis of geosmin and 2-methylisoborneol volatile compounds (S2 Table), which may promote spore dispersal by animals [8,46]. Moreover, a hydrophobic layer of chaplins may also contribute to their dispersion by limiting their loss in the anoxic depths of the soil and by mediating their adhesion to hydrophobic surfaces [47]. The induction of the expression of genes encoding gaseous vesicles (S2 Table) may also contribute to the flotation and propagation of *Streptomyces* [48]. Phage production thus seems to be part of a more general stress and/or dispersal response.

Samy is most likely activated through an SOS-independent pathway, with the precise signal (s) for Samy prophage induction yet to be determined. "NAG," "ONA," "HT," and "BM"

conditions do not share a common compound but are all associated with the induction of Samy expression. The N-acetyl-glucosamine present in the "NAG" condition (but absent in the "MM" medium) is required to induce Samy expression. This molecule is a monomer of chitin, which acts as a signal molecule and activates development and antibiotic production under poor growth conditions [49]. On the other side, ONA, HT, and its BM derivate are characterized by the presence of beef meat (**S6 Table** and **Fig 3A**). Moreover, alkalinity constitutes an additional stress signal correlated with the induction of Samy and/or enhancing phage particle stability (**Fig 3A**). Interestingly, alkaline conditions are also required for the initiation of the "explorer" phenotype in *S. venezuelae* [50]. Altogether, these observations indicate that multiple factors can promote Samy induction.

Most of our experiments were conducted under liquid growth conditions, a situation actually similar to those of most industrial bioreactors. It has long been described that the appearance of *Streptomyces* growth in liquid media (dispersed hyphae *versus* pellets or clumps) depends on the state of the initial inoculum and varies from one species to another [51]. "Fragmentation of the mycelial clumps" [52] or "pellet fragmentation" [53] in liquid medium have even been reported for some species. The biophysical parameters that may influence the type of hyphae grouping in liquid medium have been explored for a few strains [42]. Some genetic factors such as *ssgA* morphogene [52] also contribute to the morphology of *Streptomyces* mycelia in submerged cultures. Mycelium fragmentation may actually be a key contributor to strain spread in liquid medium, since most *Streptomyces*, like *S. ambofaciens* ATCC 23877, sporulate poorly or not at all during submerged growth. In this study, we demonstrate for the first time that a prophage can contribute to a similar process by promoting the formation of small and numerous clumps favoring strain dispersal and increasing by 2-logs the number of clumps/pellets able to give rise to new colonies. Samy belongs to the *Caudoviriceti* class, which are released by cell lysis. So, we propose that cell death induced by Samy in part of the colony contributes to this strain dispersal (**Fig 4H**). The fact that partial deletion of Samy (in the integrase coding region) is sufficient to abolish the phenotype supports the view that viral production is required.

Samy phage production takes place mainly between 24 h and 48 h of growth in BM medium (**Fig 3B**), leading to a final phage titer that is quite variable from one experiment to another (**Fig 3A**). Moreover, Samy production monitored by a MCP-mCherry fusion was detected only in some sections of the colony (**Figs 4E and S9C**). This suggests that the triggering of phage production in the population may be stochastic. Only a portion of the cells die as evidenced by the lack of impact on overall antibiotic production in BM liquid medium (**Fig 4G**), the high number of CFU after 4 d growth in this medium (**Fig 4D**) and the effective sporulation of the strain on solid HT medium (our personal observations). The fact that the putative immune repressor is still expressed in "HT" conditions reinforces these observations as this is characteristic of transcriptomes made on heterogeneous populations with respect to phage production, as previously reported [40]. Thus, we favor a model in which only a fraction of the cells will die, allowing the preservation of the remainder of the colony or cell aggregate. Assuming that Samy-mediated dispersion may provide a selective advantage under certain conditions (for instance, switch from BM to SFM growth condition), its production could illustrate a division of labor leading to an "altruistic" loss of fitness within a subpopulation of cells [55].

The similarities between *Streptomyces* mycelium fragmentation and biofilm dispersal have been previously highlighted [53]. Moreover, the use of phages is presented as a promising approach to control biofilms [56]. Our study highlights the possible risk of bacterial dispersal that may be associated with the (partial) induction of prophages in cellular aggregates. Phage-mediated dispersal of biofilms has also been reported in *Enterococcus faecalis* unicellular

bacteria [57]. Another prophage, in *Pseudomonas aeruginosa*, has already been shown to have an impact on bacterial biofilms by consolidating them through the contribution of extracellular DNA [58]. Our study further emphasizes the importance of considering prophages into models exploring multicellular aggregate dynamics and/or transitions to multicellular life [59].

## Materials and methods

### *Streptomyces* strains and plasmids

The *Streptomyces* strains and primers used in this study are listed in **S4** and **S5 Tables**, respectively. A CRISPR-Cas9-based approach using a single guide RNA (SBM395; **S5 Table**) targeting the Samy integrase (SAMYPH_94/ SAM23877_RS39280) was conducted in the absence of a template donor DNA using pCRISPR-Cas9 vector, as previously described [60]. The precise size of the deleted regions in 3 independent clones was determined by sequencing (**S4 Table and S8 Fig**). The gene encoding Samy MCP was cloned in fusion with a synthetic mCherry gene (Integrated DNA Technologies) within pOJ260 plasmid [61] using a Golden Gate Assembly approach (NEBridge BsaI-HFv2, New England Biolabs) with PCR products obtained with primers SBM548 to SBM555 for the inserts, and primers SBM375 and SBM376 for the vector (**S5 Table**). This plasmid, named "pOJ260-MCP-mCherry," contains homology sequences upstream and downstream of the target site as well as a spectinomycin resistance cassette in between, enabling allelic replacement by homologous recombination. The plasmid was introduced by conjugation into *S. ambofaciens*, as previously described [62]. Extraconjugants were selected on 100 μg/ml spectinomycin and 50 μg/ml nalidixic acid, and their genetic organization verified by PCR.

### Growth conditions

All media composition are described in **S6 Table**. Bacteria were grown at 30˚C. Spores were collected after growth on SFM medium. About $10^7$ spores were inoculated in 50 mL of liquid medium (in a 500-mL Duran Erlenmeyer baffled flask with a silicon stopper) and incubated in a shaking orbital agitator (200 rpm, INFORS Unitron standard). Aliquots from the cultures were collected, centrifuged, and filtered using Whatman PVDF Mini-UniPrep syringeless filters (0.2 μm). Antibacterial activity against *Micrococcus luteus* was determined, as previously described [30]. The CFUs after 4 d of growth in BM liquid medium were determined after serial dilutions in water and plating of 100 μL of dilutions on SFM medium. To analyze the presence of Samy prophage in surviving colonies, 10 isolated colonies were individually grown in TSB liquid medium and subjected to genomic DNA extraction. The presence of Samy was assessed by PCR using primers SBM385 and SBM386 (**S5 Table**). To analyze the impact of conditioned supernatants on *S. ambofaciens* growth and survival, the WT strain or its isogenic mutant Δ*Samy* clone #3 were grown in BM medium during 24 h before supplementing the medium with 0.2 μm-filtered conditioned supernatants at ratios of 1:2 or 1:5 (supernatant: culture volume). Then, the bacteria were grown for an additional 3 d before counting.

### Optical and fluorescent microscopy

For optical microscopy, after 4 d growth in BM liquid medium, 10 μL of *Streptomyces* cultures were directly mounted on glass coverslips and analyzed using an Olympus BX63 microscope. Images were acquired with a 40× oil immersion objective. The red-fluorescent microscopy was analyzed using a Leica DM6000 B microscope. Images were acquired with a 100× oil immersion objective.

## Phage quantification by qPCR

Filtered supernatants were treated with 1 U/100 µL TURBO DNAse (Invitrogen) for 1 h at 37°C and then incubated for 20 min at 80°C. Amplification was carried out on 2 µL of these treated supernatants, in a final volume of 10 µL, using the LightCycler FastStart DNA Master HybProbe kit (Roche Diagnostics) supplemented with SYBR Green. Primers used to amplify Samy prophage and a host gene are indicated in **S5 Table**. The qPCR efficiencies as well as the absolute number of amplified copies were determined using $10^2$ to $10^6$ copies of *S. ambofaciens* ATCC 23877 gDNA as standard.

## Infection assay

After 4 d growth in BM liquid medium, *S. ambofaciens* ATCC 23877 culture was centrifuged at 8,000 rpm (JA14 rotor, Beckman) for 15 min at 4°C to pellet intact bacteria and debris. The supernatant was then transferred to new centrifuge tubes and subjected to overnight centrifugation under the same condition and filtered. The pellet was resuspended and serially diluted in TBT buffer (100 mM Tris–HCl (pH 7.5); 100 mM NaCl; 10 mM $MgCl_2$) or MP Biomedicals Nutrient Broth (MNB; **S6 Table**). Approximately $10^8$ spores of target bacteria (*Streptomyces albidoflavus* J1074, *S. ambofaciens* ATCC 23877, *S. ambofaciens* DSM 40497, *S. coelicolor* A3(2), *S. venezuelae* ATCC 10712; **S4 Table**) were inoculated in 3 mL of SNA before pouring on SFM plates. Then 5 µL of serial dilutions of the phage preparation were spotted on the bacterial lawns. The plates were incubated for 1 week before imaging. In an alternative approach, 5 µL of serially diluted phage preparation was applied onto MNB agar plates supplemented with 0.5% glucose, 10 mM $MgCl_2$, and 8 mM $CaCl_2$ prior to pouring MNB soft agar (i.e., SNA; **S6 Table**) containing spores. Following this procedure, plates were incubated 24 h before imaging.

## Samy prophage detection, annotation, and classification

Synteruptor program (http://bim.i2bc.paris-saclay.fr/synteruptor/, v1.1) was used to compare the sequences of *S. ambofaciens* ATCC 23877 and *S. ambofaciens* DSM 40697 [30,32], with a minimal threshold of 20 CDSs to identify genomic islands in **Fig 1A**. The Synteruptor software's protein number axis coordinates were converted into base pair (bp) positions. PHASTER [63] was used to predict the presence of a putative prophage in the *S. ambofaciens* chromosome. *In silico* translated Samy prophage CDSs predicted in *S. ambofaciens* ATCC 28377 chromosomal sequence (CP012382, initial annotation [64]) were functionally annotated using HHPred [65] against the PDB mmCIF70 (version of January 10, 2023) and PHROGs_V4 [66] databases. Functions (known or unknown) of the nearest homologs were transferred to the queries by only retaining predictions with a probability ≥95.8%. The genome comparison between Samy and PhiC31 was performed using Easyfig software (e-value $<10^{-3}$, v2.2.5). No significant similarity with a previously annotated CDS led to a "hypothetical protein" annotation. In case of identification of a protein related to an annotated PHROG of unknown function, the annotation of the gene is "protein of unknown function" with a note indicating the related-PHROG number. This first annotation was manually curated using the more recently submitted version of *S. ambofaciens* ATCC 28377 chromosome annotation (NZ_CP012382, released April 14, 2022). The annotated genes have an identifier starting with "SAMYPH". Samy phage classification was carried out according to the guidelines set forth by the ICTV [67]. First, Samy nucleotide sequences underwent initial comparison with the "nucleotide collection (nt/nr)" database [Organism "Viruses (taxid:10239)"] using BLASTn [68] to identify potential related phage isolates at the species and/or genus taxonomic levels. Since only minimal similarities were identified this way (76% identity with a region of 429 bp of the *Gordiana*

phage Mollymur), further investigations were conducted at the family rank using VIPTree [69] and VirClust [44], 2 predicted proteome-based clustering tools. A hierarchical clustering between Samy and all the reference viral sequences of the GenomeNet/Virus-Host DB [70] was first calculated based on tBLASTx comparisons on their whole proteomes using VIPTree (https://www.genome.jp/viptree; version 4.0, default parameters). All the viral sequences of the VIPTree cluster including Samy ($n = 32$, accession numbers indicated in **S7 Fig**) were then hierarchically clustered thanks to profile-profile HMM comparisons on their whole proteomes using VirClust (https://rhea.icbm.uni-oldenburg.de/virclust/; version 2, default parameters).

## RNA preparation, sequencing, and data processing

About $2.10^6$ spores (in 5 μL) of *S. ambofaciens* ATCC23877 were spotted on a sterile cellophane covering the medium in the Petri dish. Thirty hours later, cells were harvested using a sterile cone and deposited into a tube containing 5 mL of cold ethanol. Cells were then harvested by centrifugation for 15 min at 4,000$g$ at 4°C and stored at −20°C.

RNA extraction, sequencing, and analysis were performed as previously described [30]. The reads were mapped on a genome harboring a single TIR and in terms of "Gene" and pseudogene features according to *S. ambofaciens* ATCC23877 annotation (GCF_001267885.1_ASM126788v1_genomic.gff, released April 14, 2022) and modified to include our own manual curation of Samy prophage annotation. The reference condition used to run the SARTools DESeq2-based R pipeline [71] was 24 h growth in MP5 medium. Data were analyzed with R software [72]. GO analysis g:Profiler:g:GOSt software (https://biit.cs.ut.ee/gprofiler/gost, version e109_eg56_p17_1d3191d) was performed as previously described [31]. The "mclust" R package [73] was used to identify the gene populations of Samy prophage transcriptomes, via a Gaussian finite mixture model fitted by the expectation-maximization algorithm.

## Virion purification by CsCl gradient

About $2.10^6$ spores of *S. ambofaciens* ATCC 23877 were inoculated in 2 L of BM medium. After 4 d growth, the culture was centrifuged at 8,000 rpm (JA14 rotor, Beckman) for 15 min at 4°C to pellet intact bacteria and debris. The supernatant was then transferred to new centrifuge tubes and subjected to overnight centrifugation under the same condition. The pellet was resuspended in 18 mL of TBT buffer and treated with 2 U/mL TURBO DNAse (Invitrogen) and 100 μg/mL RNAse for 1 h at 37°C. The phage was purified by isopycnic centrifugation in a CsCl discontinuous gradient for 5 h at 32,000 rpm in a SW41 rotor (Beckman) at 20°C, as previously described [74].

## Transmission electron microscopy (TEM)

Phage samples collected after 4 d growth in 1 L of HT liquid medium were concentrated for TEM observation by successive washes in ammonium acetate as previously described [40]. To eliminate debris and/or dextrin, a second set of TEM observations using the negative staining method was conducted on CsCl-purified particles produced in BM medium. Then 3 μL of sample suspension were deposited on an air glow-discharged 400 mesh copper carbon-coated grid for 1 min. Excess liquid was blotted and the grid rinsed with 2% w/v aqueous uranyl acetate. The grids were visualized at 100 kV with a Tecnai 12 Spirit TEM microscope (Thermo Fisher, New York NY, USA) equipped with a K2 Base 4k × 4k camera (Gatan, Pleasanton CA, USA).

### Virome sequencing and data processing

DNA from CsCl-purified particles was extracted twice with phenol, once with chloroform, and dialyzed overnight against 10 mM Tris–HCl (pH 7.5) and 0.1 mM EDTA. After random fragmentation, the DNA sample was paired-end sequenced using Illumina dye sequencing, as previously described [75]. Data were analyzed with the Integrative Genomics Viewer (IGV) tool [76]. STAR software [77] and *featureCounts* program [78] were used to quantify total reads along the reference genome containing only one terminal inverted repeat (TIR). The sequence coverage at termini position was identified using PhageTerm (Version 1.0.12) [79].

## Supporting information

**S1 Fig. GC content (A) and size (B) of a panel of 330 *Streptomyces* phages.**
(PDF)

**S2 Fig. Multidimensional analyses of the transcriptomes of *S. ambofaciens* ATCC 23877 grown under various conditions.**
(PDF)

**S3 Fig. Heatmaps of pSAM1 (A) and pSAM2 (B) transcriptomes in different growth conditions.**
(PDF)

**S4 Fig. Samy expression profile under the different conditions studied.**
(PDF)

**S5 Fig. Samy phage morphology and infection assays.**
(PDF)

**S6 Fig. Results of high-throughput sequencing of the double-stranded DNA virome of *S. ambofaciens* ATCC 23877 grown 4 d in BM medium.**
(PDF)

**S7 Fig. Clustering of Samy with PhiC31 and other actinophages.**
(PDF)

**S8 Fig. Schematic representation of deletions generated by a CRIPSR-Cas9 approach targeting Samy prophage integrase.**
(PDF)

**S9 Fig. Morphology of *S. ambofaciens* and *S. coelicolor* after 4 d growth in BM medium.**
(PDF)

**S10 Fig. Detection of Samy prophage within the genome of clones isolated on SFM plates after 4 d growth in BM liquid medium.**
(PDF)

**S11 Fig. Counting of colony-forming units after 4 d of growth in BM medium of *S. ambofaciens* ATCC 23877 reference strain and its isogenic Δ*Samy* #clone 3 mutant supplemented with conditioned supernatants.**
(PDF)

**S1 Table. List and characteristics of 330 *Streptomyces* phages referenced in Actinophage Database, NCBI, and/or ICTV.** The legend is detailed in the "Readme" sheet (to download separately online).
(XLSX)

**S2 Table. Results from the OSMAC-RNAseq approach conducted in *S. ambofaciens* ATCC 23877.** The legend is detailed in the "Readme" sheet (to download separately online).
(XLSX)

**S3 Table. List of Samy genes overexpressed in non- or poorly inducing conditions.** The legend is detailed in the "Readme" sheet (to download separately online).
(XLSX)

**S4 Table. Strains used in this study.**
(PDF)

**S5 Table. Primers used in this study.**
(PDF)

**S6 Table. Media used in this study.**
(PDF)

**S1 Data. Data values for all figures within this study.**
(XLSX)

**S2 Data. Scripts used for RNA-seq analysis and generate the figures of this study.**
(HTML)

**S3 Data. Statistical report of the RNA-seq analysis.**
(HTML)

**S1 Raw Image. Raw image for S10 Fig.**
(JPG)

## Acknowledgments

We would like to thank Julien Lossouarn (MICALIS, INRAE, Jouy-en-Josas), Lambda-like viruses study group member of the ICTV, for his valuable help in annotating phage genes, submitting Samy's genome and in the classification of the Samy phage. Additionally, we express our appreciation to the team at PhagesDB.org for their assistance in the submission and clustering of the Samy sequence in the Actinophage Database. We also thank Jean-Luc Pernodet and Frederic Boccard for their precious support. We also would like to thank Christine Longin (MIMA2 platform) for her help with TEM observations. We thank Soumaya Najah and Corinne Saulnier for their technical help. We are grateful to Vinciane Regnier and Logan Greibill for valuable training on the Olympus BX63 microscope. We acknowledge the sequencing and bioinformatics expertise of the I2BC High-throughput sequencing facility, supported by *France Génomique* (funded by the French National Program *Investissement d'Avenir* ANR-10-INBS-09). We also thank Pascaline Tirand for her daily help.

## Author Contributions

**Conceptualization:** Virginia S. Lioy, Stéphanie G. Bury-Moné.

**Data curation:** François Lecointe, Stéphanie G. Bury-Moné.

**Formal analysis:** Hoda Jaffal, François Lecointe, Stéphanie G. Bury-Moné.

**Funding acquisition:** Sylvie Lautru, Virginia S. Lioy, Stéphanie G. Bury-Moné.

**Investigation:** Hoda Jaffal, Mounia Kortebi, Pauline Misson, Paulo Tavares, Hervé Leh, Stéphanie G. Bury-Moné.

**Methodology:** Paulo Tavares, Malika Ouldali.

**Project administration:** Stéphanie G. Bury-Moné.

**Supervision:** Sylvie Lautru, Virginia S. Lioy, François Lecointe, Stéphanie G. Bury-Moné.

**Validation:** Hoda Jaffal.

**Visualization:** Mounia Kortebi.

**Writing – original draft:** François Lecointe, Stéphanie G. Bury-Moné.

**Writing – review & editing:** Hoda Jaffal, Mounia Kortebi, Pauline Misson, Paulo Tavares, Malika Ouldali, Hervé Leh, Sylvie Lautru, Virginia S. Lioy.

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
