## [Editor Report · Decision Letter 0]

12 Jan 2024

Dear Dr Bury-Moné, 

Thank you for submitting your manuscript entitled "Prophage induction can facilitate the in vitro dispersal of multicellular bacteria" for consideration as a Research Article by PLOS Biology.

Your manuscript has now been evaluated by the PLOS Biology editorial staff, as well as by an academic editor with relevant expertise, and I am writing to let you know that we would like to send your submission out for external peer review.

IMPORTANT: Your paper is submitted as a regular Research Article. However, after discussion with the Academic Editor, we think that your study would be better considered as a Short Report (https://journals.plos.org/plosbiology/s/what-we-publish#loc-short-reports). Very little re-formatting is required, but we would need you to reduce the number of Figure down to 4. You could do this either by combining multiple Figures or by moving some of the material in the existing main Figures to the supplement. Please do this, and select "Short Reports" as the article type, when uploading your additional metadata (see next paragraph).

Once your full submission is complete, your paper will undergo a series of checks in preparation for peer review. After your manuscript has passed the checks it will be sent out for review. To provide the metadata for your submission, please Login to Editorial Manager (https://www.editorialmanager.com/pbiology) within four working days, i.e. by Jan 18 2024 11:59PM.

Kind regards,

Melissa

Melissa Vazquez Hernandez, Ph.D.

Associate Editor

PLOS Biology

---

## [Decision Letter · Decision Letter 1]

16 Feb 2024

Dear Dr Bury-Moné,

Thank you for your patience while your manuscript "Prophage induction can facilitate the in vitro dispersal of multicellular bacteria" was peer-reviewed at PLOS Biology. It has now been evaluated by the PLOS Biology editors, an Academic Editor with relevant expertise, and by three independent reviewers, one of whom signed their report and is Dennis Claessen.

In light of the reviews, which you will find at the end of this email, we would like to invite you to revise the work to thoroughly address the reviewers' reports. As you will see below, although the reviewers see the potential interest in your work, there are still concerns that should be addressed. Overall, there is sufficient interest for us to consider the study further, but some revision will be necessary for publication in PLOS Biology. Reviewer #1 suggests that the taxonomy and characteristics of the prophage should be expanded, and reviewers #2 and #3 would like to see more in depth mechanistic insights, such as the mechanism of induction. Additionally, during cross-comments (here paraphrased), reviewers suggested to evaluate if NAG can be an inducible signal. If verified, this finding could be linked to programmed cell death. Additional suggestions include fluorescence in situ hybridisation or live/dead staining to evaluate if prophage induction leads to lysis of fragments in the mycelium. You should also consider previous reports of prophages in actinobacteria such as doi: 10.1016/j.xgen.2022.100213 and doi: 10.1038/s41598-023-30829-z.

IMPORTANT: as this is submitted as a Short Report, we do not require you to decipher fully the mechanism of induction. HOWEVER, for publication in PLOS Biology, we do require additional experimental and phylogenetic analysis to better characterise this phage and its induction, following the recommendations of the reviewers.

Given the extent of revision needed, we cannot make a decision about publication until we have seen the revised manuscript and your response to the reviewers' comments. Your revised manuscript may be sent for further evaluation by all or a subset of the reviewers.

**IMPORTANT - SUBMITTING YOUR REVISION**

*Re-submission Checklist*

*Published Peer Review*

*PLOS Data Policy*

*Blot and Gel Data Policy*

Sincerely,

Melissa

Melissa Vazquez Hernandez, Ph.D.

Associate Editor

PLOS Biology

REVIEWER's COMMENTS

Reviewer #1: Yes: Dennis Claessen

Streptomyces bacteria are well-known for their production of specialized metabolites vital in medicine and agriculture, and their interactions with bacteriophages remain poorly understood. This study focused on the 'Samy' prophage in Streptomyces ambofaciens and revealed its simultaneous production with activation of other genetic elements and its ability to increase bacterial dispersal under stress. This research sheds new light on the role of bacteriophage infections in the dynamics of multicellular aggregates.

The manuscript is well-written, and the authors bring a new element to the life cycle of these bacteria under influence of stress. The experiments have been weel designed and carried out, and the major conclusions are valid based on the presented data. I have a few suggestions to further improve this story, which overall I am very much impressed with:

- In line 47, the authors imply that all identified Streptomyces phages are double stranded DNA viruses. But are really "all" Streptomyces phages double stranded DNA viruses? I think this sentence should be rephrases to avoid ambiguity. 

- In the introduction, nothing is mentioned on how a bacteriophage infects a multicellular bacteria. Most research is performed on unicellular model organisms, but can the authors speculate on how phages infect a multicellular bacteria? For example, can a phage travel through the hyphae? If one part of a mycelium is infected, will the whole mycelial aggregate lyse? It would be nice to speculate a bit on this infection of multicellular bacteria in the introduction. 

- When talking about Streptomyces having a viral genetic reservoir in line 64, the authors refer to a paper that identified prophages in seven Streptomyces strains, including S. coelicolor to my understanding. However, in line 314, the authors state that S. coelicolor does not have a predicted complete prophage nor Samy. Could the others comment on this? Could it be that the used conditions may not activate the prophage?

In addition, this paper (https://doi.org/10.1016/j.xgen.2022.100213) might be a nice reference to show that almost 50% of all actinobacteria have a predicted prophage. Please make the distinction between bioinformatics suggesting a prophage and experimentally verifying the computational results.

- In line 72, the authors say that prophages are produced, but I would suggest to change the word to "induce" under these conditions. 

- The supplementary figures are not in the correct order. So, Figure S2 should actually be S1 in the text etc. Please check this for the whole manuscript. 

- I think the taxonomy and characteristics of Samy should be elaborated. In the material and methods from line 511-522 the authors speak of infection assays, but did you also perform plaque assays? I did not understand why these assays were grown for a week, while the common Streptomyces plaque assays are incubated for 1 day at 30 degrees with really pretty results. This could also explain why the authors did not see any clear infections of Samy on the tested streptomyces strains. I would suggest to repeat this experiment with DNB or GYM plaque assays and perform a double agar overlay assay. Can Samy infect the S. ambofaciens strains without the prophage and the DSMZ strain of S. ambofaciens? If yes or not, please also explain in the text in line 286-292. Next to that, is Samy classified as a new species? Since line 71 says it is a novel phage, please also explain to which genus and family this phage belongs. Since novel phages are classified between >75 and <95 genome similarity according to ICTV , also provide data to prove that this actually is a novel phage, which can be done with VIRIDIC for example. Lastly in figure 1, why is Samy compared to phiC31. This is a common and well known streptomyces phage, but is this also the phage that belong in the same genus as Samy? Please compare Samy to phages in the same genus in figure 1C. 

- Did the authors test if the CFU's that could be counted after Samy induction still have Samy in there genome? Or are these surviving parts of the hyphae that did not undergo lysis and therefore Samy induction? A bit more speculation on how the authors think that Samy causes these smaller fragments without killing the whole population would benefit the discussion greatly. In line 442 the authors propose that Samy induction in a part of the colony contributes to strain dispersal, but why is Samy only induced in a part of the colony? Please elaborate. 

- In Figure 2C, the difference between plates and liquid should be made more clear. Perhaps put a small line between MM and MP5 to make this distinction more visual. 

- The microscopy images in Figure 4C and supplementary figure 8A and B are very difficult to see. Is it possible to improve these pictures? 

Reviewer #2: 

The authors report an inducible prophage (Samy) found in the genome of Streptomyces ambofaciens. Infectivity of the induced phage virions was confirmed on strains lacking the integrated prophages. The authors performed a comprehensive transcriptome analysis to identify media on which Samy is induced. Finally, they show that prophage induction boosters the dispersal of Streptomyces, leading to multicellular aggregates of smaller size. Therefore, the study provides interesting - yet not mechanistic - insights into the impact of prophage induction on the lifestyle of multicellular bacteria.

Comments

General comments:

1) My major concern is that the study lacks any conclusive mechanistic insights or looks further into the broader relevance of this phenomenon (prophage induction in other Streptomyces strains)

2) The impact of prophage induction on multicellular aggregates is interesting, yet not surprising. I believe that induction of Samy is highly heterogeneous and only happens in parts of the mycelium, leading to fragmentation. A visualization of this induction pattern, e.g. by FISH using phage-specific probes, would be highly interesting.

3) The authors perform a comprehensive screening of different media conditions leading to prophage induction. However, they do not really nail it down to specific factors leading to induction. They find that N-acetyl-glucosamine is present in the NAG condition and might be the responsible factor. This would be straightforward to check by adding it to non-inducing conditions.

4) Noteably, the authors do not comment on the phylogenetic and taxonomic classification of the newly identified phage. Strikingly, no closest relative could be determined by blast search using the complete genome. Using single proteins, e.g. DNA polymerase/primase, terminase large subunit, major head protein, tail length tape measure protein, endolysin, tail fiber protein or immunity repressor it is interesting to note that these share highest similarities with phages infecting very different species. Best hits include Streptomyces, Gordonia, Mycobacterium and Arthrobacter phages. Therefore, it would be very interesting to test, if Samy could indeed infect any of these and at least discuss these circumstances.

Specific comments:

Line 65: A more general, systematic analysis of prophage abundance in Streptomyces genomes is provided by Sharma et al., 2023 (https://doi.org/10.1038/s41598-023-30829-z). In contrast to only 7 strains, here a total of 314 Streptomyces genomes were analysed and resulted in 62% containing prophages.

Line 81: The authors directly jump into S. ambofaciens, without explaining why this actually is an interesting model or prophage to study. 

Fig 2B: I think it is difficult to compare gene expression between solid and liquid growth plus using two different time points for comparison. In this experiment, strongest induction was observed on solid media. Is induction on solid media in general more pronounced? But there no direct comparison on liquid and solid media for any of the media/conditions, which I found surprising. 

Line 286 ff: Can Samy also lysogenize S. lividans? Or if the infection of this strain is strictly lytic. 

Reviewer #3: Jaffal and colleagues identified a prophage in Streptomyces ambofaciens that upon induction leads to dispersal of bacteria. The phage, named "Samy", was identified through differences in GC content (66%) and annotated using PHASTER. Through different growth conditions (e.g. different media, alkaline growth conditions, different growth phases), the authors concluded that Samy is expressed during a general stress response, although this seems to be independent of the SOS response (with the signals for induction yet to be determined). Growth data indicate a complex role of Samy in influencing Streptomyces growth dynamics, particularly under stress conditions, by preventing large cell aggregate formation and promoting bacterial dispersal, without adversely affecting antibiotic production. All of this indicates that phage production is part of a more general stress and dispersal response of S. ambofaciens but the role of this phage in growth, division, adaptation to environmental niches, production of antibiotics and other compounds, has not been studied. 

In summary, the authors present interesting observations and describe a novel prophage together with its genetic and metabolic pathways involved. However, the precise mechanisms and signals leading to the production and release of phages, and the biologic role this mechanism plays during the bacterial life cycle are not explai

---

## [Decision Letter · Decision Letter 2]

15 Jun 2024

Dear Dr Bury-Moné,

Thank you for your patience while we considered your revised manuscript "Prophage induction can facilitate the in vitro dispersal of multicellular bacteria" for publication as a Short Reports at PLOS Biology. This revised version of your manuscript has been evaluated by the PLOS Biology editors, the Academic Editor and the original reviewers.

Based on the reviews, we are likely to accept this manuscript for publication, provided you satisfactorily address the remaining editorial points. Please make sure to address the following data and other policy-related requests.

a) We would like to suggest the following modification to the title:

"Prophage induction can facilitate the in vitro dispersal of multicellular Streptomyces structures"

Please supply the numerical values either in the a supplementary file or as a permanent DOI’d deposition for the following figures:

Figure 1A, 2AB, 3AB, 4ADFG, S1AB, S2C, S2ABC, S4, S11

c) Please cite the location of the data clearly in all relevant main and supplementary Figure legends, e.g. “The data underlying this Figure can be found in S1 Data” or “The data underlying this Figure can be found in https://doi.org/10.5281/zenodo.XXXXX”

d) We require the original, uncropped and minimally adjusted images supporting all blot and gel results reported in the Figures S10

We will require these files before a manuscript can be accepted so please prepare and upload them now. Please carefully read our guidelines for how to prepare and upload this data: https://journals.plos.org/plosbiology/s/figures#loc-blot-and-gel-reporting-requirements

e) Please provide the tree files for Figures S2B and S7AB

f) Please ensure that your Data Statement in the submission system accurately describes where your data can be found and is in final format, as it will be published as written there.

g) Per journal policy, if you have generated any custom code during the curse of this investigation, please make it available without restrictions upon publication. Please ensure that the code is sufficiently well documented and reusable, and that your Data Statement in the Editorial Manager submission system accurately describes where your code can be found.

We expect to receive your revised manuscript within two weeks. 

*Published Peer Review History*

*Press*

Sincerely,

Melissa

Melissa Vazquez Hernandez, Ph.D.

Associate Editor

PLOS Biology

REVIEWERS' COMMENTS

Reviewer #1: 

The authors have properly addressed all raised concerns and I would like to congratulate the authors with their great work!

Reviewer #2: 

The authors submit a revised version of their manuscript. They added further information regarding prophage distribution in Streptomyces strains (citing the relevant literature) and on taxonomy and characteristics of the Samy prophage. Their analysis identifies Samy as a singleton phage with no close relative among sequences Streptomyces phages - which is frequently the case for prophages in actinobacterial genomes; they often do not match with kown phage isolate.

To visualize induction of Samy in the mycelium, they have included microscopy of a strain carrying a MCP-mCherry fusion of the phage. The picuture (Fig. 4E) confirms indeed that inductions only happens in some sections of the mycelium and are a valuable addition to the manuscript. However, it would have been interesting to analyse this strain on the different media leading to diffenrent induction patterns of the prophage to see whether this would align well. Further, a quantification would be nice, but I believe that this can be difficult for mycelial structures adding a third dimension in the pictures...

Figure R4 presents further data on the analysis of NAG's effect on prophage induction. These data show contradictory patterns (+/-NAG) across different media, making it currently impossible to draw definitive conclusions about the impact of media composition and cultivation conditions on prophage activation. 

Therefore, as outlined in my previous report, I find this an interesting study. The fact that spontaneous prophage induction contributes to the dispersal of multicellular bacteria is certainly a noteworthy observation. Whether this concepts would also be more widespread in other Streptomyces species is certainly interesting to adress.

Due to the complexity of the microbial system, the study remains on a descriptive level and the underlying trigger remains elusive. 

Reviewer #3: 

The authors have improved the manuscript and have added additional information and answered some of the issues raised by the reviewers

---

## [Editor Report · Decision Letter 3]

28 Jun 2024

Dear Stéphanie,

Thank you for the submission of your revised Short Reports "Prophage induction can facilitate the in vitro dispersal of multicellular Streptomyces structures" for publication in PLOS Biology. On behalf of my colleagues and the Academic Editor, Jeremy Barr, I am pleased to say that we can in principle accept your manuscript for publication, provided you address any remaining formatting and reporting issues. These will be detailed in an email you should receive within 2-3 business days from our colleagues in the journal operations team; no action is required from you until then. Please note that we will not be able to formally accept your manuscript and schedule it for publication until you have completed any requested changes.

PRESS

Sincerely, 

Melissa

Melissa Vazquez Hernandez, Ph.D., Ph.D.

Associate Editor

PLOS Biology
